# Effects of the Marek’s Disease Vaccine on the Performance, Meat Yield, and Incidence of Woody Breast Myopathy in Ross 708 Broilers When Administered Alone or in Conjunction with *In ovo* and Dietary Supplemental 25-Hydroxycholecalciferol

**DOI:** 10.3390/ani14091308

**Published:** 2024-04-26

**Authors:** Seyed Abolghasem Fatemi, Ayoub Mousstaaid, Christopher J. Williams, Joshua Deines, Sabin Poudel, Ishab Poudel, Elianna Rice Walters, April Waguespack Levy, Edgar David Peebles

**Affiliations:** 1Department of Poultry Science, Mississippi State University, Mississippi State, MS 39762, USA; am4768@msstate.edu (A.M.); sp2186@msstate.edu (S.P.); ishabpoudel@gmail.com (I.P.); erw263@msstate.edu (E.R.W.); d.peebles@msstate.edu (E.D.P.); 2Zoetis Animal Health, Research Triangle Park, Durham, NC 27703, USA; cjw704353@gmail.com (C.J.W.); joshua.deines@zoetis.com (J.D.); 3DSM Nutritional Products, Parsippany, NJ 07054, USA; april.levy@dsm.com

**Keywords:** 25-hydroxyvitamin D_3_, broiler, *in ovo* administration, Marek’s disease vaccine, performance

## Abstract

**Simple Summary:**

The second metabolite of vitamin D, 25-hydroxyvitamin D_3_ (25OHD_3_), has previously shown promising results on the live performance and meat yield of broilers when it was administered *in ovo* or in their diet. However, the effects of 25OHD_3_ on the posthatch performance of broilers have not been tested in combination with the *in ovo* administration of the Marek’s disease vaccine (MDV). Therefore, the aim of the current research was to investigate the effects of *in ovo* and dietary sources of 25OHD_3_ in conjunction with the *in ovo* delivery of the MVD on the broiler meat yield, live performance, and incidence of woody breast myopathy (WBM). In this study, it was shown that both 25OHD_3_ sources increased the meat yield and improved the live performance variables of broilers with no measurable negative effects on WBM scoring. It is worth mentioning that the dietary source of 25OHD_3_ had greater effects on breast meat yield and posthatch performance throughout the rearing period when compared to its *in ovo* administration. In conclusion, both the *in ovo* and dietary administration of 25OHD_3_ can be used in combination with the *in ovo* delivery of the MDV in order to enhance its efficacy on broiler posthatch production.

**Abstract:**

The effects of the Marek’s disease vaccine (MDV) on the live performance, breast meat yield, and incidence of woody breast myopathy (WBM) of Ross 708 broilers were investigated when administered alone or in conjunction with *in ovo* and dietary supplemental 25-hydroxycholecalciferol (25OHD_3_). At 18 d of incubation (doi), four *in ovo* injection treatments were randomly assigned to live embryonated Ross 708 broiler hatching eggs: (1) non-injected; (2) commercial MDV alone; or MDV containing either (3) 1.2 or (4) 2.4 μg of 25OHD_3_. An Inovoject multi-egg injector was used to inject a 50 μL solution volume into each egg. The birds were provided a commercial diet that contained 250 IU of cholecalciferol/kg of feed (control) or a commercial diet that was supplemented with an additional 2760 IU of 25OHD_3_/kg of feed (HyD-diet). In the growout period, 14 male broilers were placed in each of 48 floor pens resulting 6 replicated pens per *in ovo* x dietary treatment combination. Live performance variable were measured at each dietary phases from 0 to 14, 15 to 28, and 29 to 40 d of age (doa). At 14 and 40 doa, pectoralis major (P. major) and pectoralis minor (P. minor) muscles were determined for one bird within each of the six replicate pens. At 41 doa, WBM incidence was determined. No significant main or interaction effects occurred for WBM among the dietary or *in ovo* injection treatments. However, in response to *in ovo* 25OHD_3_ supplementation, BW and BWG in the 29 to 40 doa period and BWG and FCR in the 0 to 40 doa period improved. In addition, at 40 and 41 doa, breast meat yield increased in response to *in ovo* and dietary 25OHD_3_ supplementation. Future research is needed to determine the possible reasons that may have been involved in the aforementioned improvements.

## 1. Introduction

It is well observed that an important and pragmatic innovation in the past 3 decades that has shown a significant influence on the poultry industry is *in ovo* injection technology. It has been designed to provide a stress-free, earlier, faster, and uniform delivery of vaccines for the protection of broilers, and it has been emerged as an alternative approach to the posthatch vaccination of chickens, particularly in broilers [1,2,3,4]. *In ovo* injection is used for the direct administration of particular nutrients or vaccines in the amnion of embryos between 17.5 and 19.25 d of incubation (**doi**) [2,3,4]. Furthermore, the 18 doi *in ovo* administration of several vaccines for *Escherichia Coli* [5], *Mycoplasma galicepticum* [6], infectious bursa disease [7] and Marek’s disease [8] and various supplemental nutrients including vitamins, minerals, and carbohydrates [1,2,9,10,11] have been shown to promote not only hatchling immunity but also to improve hatchability and posthatch performance. Although the use of various vaccine has been well established in US broiler hatcheries, the use of any nutrients including vitamins minerals, proteins, amino acids, and organic acids has not been commercially used. One major reason could be due to the lack of their combined effects with any commercial vaccines and the fact that their subsequent effects on the broiler production variables have not been investigated. Currently, among the aforementioned vaccines, the *in ovo* injection of the Marek’s Disease vaccine (**MDV;** turkey herpesvirus) is widely used in U.S. commercial broiler hatcheries in order to enhance early immunity [1,2,10,11,12]. It is well documented that the *in ovo* injection of the MDV provides over 90% protection when it is administrated via the amnion or body proper [2]. Additionally, it has resulted in an enhancement of small intestine morphology [13] and meat yield [14], as well as the expression of genes linked to humoral immunity [8].

Vitamin D is a fat-soluble vitamin that has a wide range of biological functions in chickens including bone formation and development [15,16], immune system regulation [17,18,19,20,21], the intestinal absorption of Ca and phosphorous [22], small intestine histomorphology [18,19,23,24], and muscle development [25,26,27,28]. Cholecalciferol (**D_3_**) is the first metabolite of vitamin D after its absorption in the gut. Upon being bound to vitamin D-binding proteins, it is transported to the liver where its first hydroxylation step takes place to convert it to 25-hydroxycholecalciferol (**25OHD_3_**) via 25-hydroxylase [29]. The second hydroxylation step, which makes vitamin D an active hormone, occurs in the renal cells by the action of 1α-hydroxylase, where 25OHD_3_ is converted to *1,25-dihydroxyvitamin D_3_* (**1,25-(OH)_2_-D_3_**), the active form of vitamin D. In contrast to D_3_, dietary 25OHD_3_ supplied in the feed at a level of 69 mg/kg (equivalent to 2760 IU/kg), has been shown to increase breast [26] and leg [27] meat yields and to improve small intestine morphology [23,24] and adaptive and innate immunity [17,20,21,30,31,32]. It is suggested that the beneficial results in response to 25OHD_3_ can be linked to its longer half-life in comparison to that of D_3_ (3 wk vs. 15 h) [33,34], its ability to stimulate an increase in Ca and phosphorus absorption in the small intestine [22], and its greater storage in muscle tissue [35]. For many decades, the poultry industry has improved meat yield and production efficiency as well as disease control using intensive genetic selection [36]. However, this rapid growth rate has been shown to be linked to increased metabolic diseases and abnormalities in breast fillets exhibiting woody breast myopathy (WBM) as a result of an increase in myodegeneration, lipidosis, fibrosis, and oxidative stress and a decrease in protein synthesis in pectoralis (P.) major [37]. It is estimated that some levels of WMB can be detected in approximately 9% of the breast fillets of larger broilers (2.72–4.53 kg) [38]. It is worth-mentioning that WMB not only negatively affects meat quality but also has negative effects on production profits. This is largely due to the fact that severe WBM breasts are not allowed to be sold and that breasts with moderate WBM scores are usually sold at half price [37,39]. It has been suggested that the nutritional agents that can increase protein synthesis and reduce inflammation in fillets can be used to reduce WBM incidence [37,39]. Therefore, vitamin D_3_ sources that stimulate immunocompetent and muscle synthesis may lower WBM incidence.

Compared to a non-injected control, the *in ovo* feeding of 100 μL of 1.2 to 3.6 μg of 25OHD_3_ suspended in olive oil showed no beneficial results on the hatchability of injected live embryonated eggs (**HI**) and the bone quality of broilers [40]. Additionally, greater posthatch impacts of a water-soluble source of *in ovo*-injected 25OHD_3_ were reported in broilers in comparison to those that belonged to D_3_ alone and diluent-injected control groups under commercial conditions. The *in ovo* injection of 2.4 µg of 25OHD_3_ has also been shown to increase broiler hatchling quality and serum 25OHD_3_ concentrations [41] and to improve subsequent breast meat yield [42,43] and posthatch live performance [41,42,43,44], anti-inflammatory response [23,42], humoral immunity [22], and small intestine histomorphology [22]. Moreover, compared to non-injected and sterile water-injected treatments, the *in ovo* injection of 8 μg/mL of 25OHD_3_ suspended in ethanol resulted in an increase in the HI, posthatch body weight (BW), tibial weight, tibial length, tibial diameter, and immune organs weight of hatchlings when it was administered at 17.5 doi [45]. More recently, the amniotic *in ovo* injection of 2.4 μg of 25OHD_3_ has been shown to increase the breast and leg meat yield [46], decrease the inflammatory reaction [23], and to improve the posthatch live performance [46], small intestine morphology [23], and expression of genes associated with D_3_ activity [24] in Ross 708 broilers subjected to a coccidiosis infection. Furthermore, Fatemi et al. [47] demonstrated that low (0.6 μg) and high (2.4 μg) doses of 25OHD_3_ have negligible negative effects on MDV cell survival. Additionally, the 1.2 and 2.4 μg doses of 25OHD_3_ in combination with MDV have been observed to increase hatchling BW and the expression of genes linked to immunity and vitamin D activity [48]. Thus, these doses may be appropriate candidates for *in ovo* injection in combination with MDV in order to determine their subsequent effects on various posthatch variables. The effects of the administration of 25OHD_3_ in conjunction with MDV on the posthatch performance, meat yield and quality of broilers were not used in previous investigations. Therefore, the current objectives were to determine the effects of the *in ovo* injection of various doses of 25OHD_3_ in combination with the MDV on the posthatch live performance, meat yield, and quality of Ross 708 broilers.

## 2. Materials and Methods

### 2.1. Egg Incubation and Experimental Design

From 35-week-old commercial Ross 708 broiler breeder hens, fertile eggs were collected and stored for 24 h under recommended conditions (12.8 °C and 10.4 °C dry and wet bulb temperatures) for 24 h [5,41]. Twelve replicate groups (blocks), each containing 40 eggs, were assigned to each of the 4 *in ovo* injection treatment groups (1920 total eggs) in a single-stage setter/hatcher incubator (Chick Master Incubator Company, Medina, OH, USA). The setter phase was set at 37.5 °C dry bulb and 29.0 °C wet bulb temperatures and the hatcher at 36.9 °C dry bulb and 29.9 °C wet bulb temperatures. All eggs were candled at 12 and 18 doi in order to remove infertile eggs or those that contained dead embryos based on the method demonstrated by Ernst et al. [49]. In addition, the mean percentage egg weight loss (**PEWL**) for each treatment replicate group of eggs was determined between 0 and 12, 12 and 18, and 0 and 18 doi in order to confirm that a uniform incubational condition was experienced for all treatment groups. At 18 doi, 50 μL solution volumes of each pre-specified treatment were injected into eggs using a Zoetis Inovoject m (Zoetis Animal Health, Research Triangle Park, NC, USA) multi-egg injection machine. The *in ovo* injection treatments were (1) non-injected; (2) commercial MDV alone; or MDVs that contained (3) 1.2 μg of 25OHD_3_ (**MDV + 25OHD_3_-1.2**) or (4) 2.4 μg of 25OHD_3_ (**MDV + 25OHD_3_-2.4**). The form and source of 25OHD_3_ used in this study (ROVIMIX^®^HY-D^®^1.25%; DSM Nutritional Products, Inc., Parsippany, NJ, USA) was the same as that used by Fatemi et al. [47,48]. All *in ovo* injection solutions were also prepared and injected according to the procedures of Fatemi et al. [47,48]. In addition, one live embryonated egg from each of the 4 treatment groups on each of the 12 incubator tray levels (48 total eggs) was selected for embryo staging analysis including an embryo development stage score (**ES**) and site of injection in accordance to the method described by Fatemi et al. [48]. At 21 doi, all hatch variables including the hatchability of set eggs (**HS**), HI, hatchling BW, and hatch residue were determined. Hatch residue analysis was performed on eggs after candling between 18 to 21 doi, according to the procedure described by Fatemi et al. [48], to ensure that only live embrocated eggs were included until hatch. This analysis included late embryonic mortality (**LEM**), prior piped embryonic mortality (**PPM**), post piped embryonic mortality (**PEM**), and hatchling mortality.

At hatch (21 doi), all chicks were feather-sexed to select for male broilers in their pre-specified treatment, and then 13 male broilers were placed at a 0.62 m^2^/bird stocking density in each of 48 floor pens containing used litter. The dietary treatments that were assigned included (1) a commercial diet supplemented with an additional 250 IU of vitamin D_3_/kg of feed (control) or (2) a commercial diet plus 2760 IU of 25OHD_3/_kg of feed (**Hy-D** diet). The experimental design resulted in a total of 8 treatment groups (2 dietary treatments x 4 *in ovo* treatments). There were 6 replicate pens per treatment in a randomized complete block design. Chicks had *ad libitum* access to feed and fresh water. The starter diet was fed from 0 to 14 d of age (**doa**), the grower diet was fed from 15 to 21 doa, and the finisher diet was fed as pellets from 22 to 40 doa. Birds were processed at 41 doa at Mississippi State University poultry farm processing plant. All diets were Mississippi State University basal corn–soybean diet formulations that met the Ross 708 commercial guidelines (Table 1) [42,43,46,50]. Diets were fed as crumbles from 0 to 14 d of age and then as pellets from 15 to 40 doa. Three dietary phases were considered as follows: starter from 0 to 14 doa; grower from 15 to 28 doa; and finisher from 29 to 40 doa. The analyzed dietary D_3_ and 25OHD_3_ levels in the starter, grower, and finisher phases are shown in Table 2. The actual values for D_3_ ranged from 80 to 110% of the formulated values for D_3_-containing diets, and the actual 25OHD_3_ levels ranged from 85 to 101% of the formulated values for 25OHD_3_-containing diets.

### 2.2. Live Performance

For each pen, live performance variables including mean bird BW, feed intake (**FI**; g/bird), average daily FI (**ADFI;** g/bird), BW gain (**BWG**), average daily BW gain (**ADG**), and feed conversion ratio (**FCR**; g feed/g gain) were determined for each dietary phase. These data as well as the FCR were adjusted for bird mortality, and the percentage total mortality was also calculated for the overall 0 to 40 doa period.

### 2.3. Meat Yield and Woody Breast Myopathy Scoring

At 14 and 40 doa, one bird from each of the 6 replicate pens per treatment (48 total birds) were randomly selected and individually weighed, and the relative weights of their P. major and P. minor muscles relative to the total BW were determined. The total breast weight (the sum of P. major and P. minor weights) was also calculated. The approximately 7 remaining birds in each pen were processed at 41 doa. Prior to slaughter, the birds did not have access to feed or water for at least 12 h. the birds were processed according to the procedure of Fatemi et al. [43]. At processing, whole carcass and P. major, P. minor, drumstick, thigh, wing, and fad pad parts weights and yields (portion weight as a % of carcass weight) were determined. At 41 doa, the P. major samples were scored for the incidence of WBM according to the method described by Fatemi et al. [43] and Tijare et al. [37].

### 2.4. Statistical Analysis

In the incubation and hatch phases of the experiment, incubator tray levels contained each of the 5 *in ovo* treatments. In addition, for the posthatch period, the experimental unit was the floor pen. A randomized complete block experimental design was used for both the incubational and growout periods. With all *in ovo* injection treatments randomly represented on each of the 12 incubator tray levels (blocks), the incubator tray level was the blocking factor, and with both the dietary and *in ovo* injection treatments (2 × 4) being randomly represented in each of 6 pens, a group of 6 pens was the blocking factor. A one-way ANOVA was used to test for the effects of the 4 *in ovo* injection treatments on the incubation and hatch data.

A two-way ANOVA was used to analyze the performance and meat yield data with a 4 × 2 factorial arrangement of treatments to test for the main and interactive effects of the 4 *in ovo* injection treatments and 2 dietary treatments. The following model was performed for the analysis of the performance and meat yield data:Y_ijk_ = μ + B_i_+ I_j_ + D_k_ + (ID)_jk_ + E_ijk_,
where μ is the population mean; B_i_ is the block factor (i = 1 to 2); I_k_ is the effect of the *in ovo* injection treatment (k = 1 to 4); D_i_ is the effect of each dietary treatment (j = 1 to 2); (ID)_ij_ is the interaction of each dietary treatment with the *in ovo* injection treatment; and E_ij_ is the residual error.

All data were analyzed using the general linear mixed models (PROC GLIMMIX) of SAS 9.4© [51], and Fisher’s protected least significant difference analysis was performed for the separations of the treatment means [52], with treatment differences considered significant at *p* ≤ 0.05.

Furthermore, differences among the mean WBM scores were analyzed using the nonparametric models procedure (PROC NPAR1WAY) and PROC GLIMMIX of SAS 9.4© [51]. Differences among the means were considered to be significant at *p* ≤ 0.05.

## 3. Results

### 3.1. Hatch Variables

The mean ES scores of the live embryonated eggs at 18 doi were 2.33, 2.50, 2.17, and 2.33 for the non-injected, MDV alone, MDV + 25OHD_3_-1.2, and MDV + 25OHD_3_-2.4 treatments, respectively. Regardless of treatment effect, the average ES score was 2.33 (S.D. = 0.816), which indicated that the embryos were positioned prior to piping and with their heads located under the right wing. Furthermore, the site of injection evaluations showed that 4.17, 91.67, and 4.17% of the eggs were vaccinated, respectively, in the air cell, amnion, and body proper. There were no significant differences among the *in ovo* treatments for egg weight, PEWL at all time periods, HS, HI, PPM, PEM, hatchling mortality, and hatchling BW. However, LEM in the MDV-alone and MDV + 25OHD_3_-2.4 treatments was significantly higher than that in the non-injected treatment group, with that in the MDV + 25OHD_3_-1.2 treatment being intermediate (Table 3).

### 3.2. Live Performance

There were no significant interactive effects between the *in ovo* and dietary treatments for all the live performance variables within and across the starter, grower, and finisher phases of the rearing period (Table 4). Also, there were no significant main or interactive effects of the *in ovo* and dietary treatments on total bird mortality between 0 and 40 doa (Table 4). From 0 to 14, 15 to 28, 29 to 40, and 0 to 40 doa, various live performance variables were improved in broilers fed supplemental dietary Hy-D in comparison to those fed an unsupplemented commercial diet. More specifically, BW, BWG, ADG, FI, and ADF from 0 to 14, 15 to 28, and 29 to 40 doa were significantly higher in birds fed the Hy-D-supplemented diet in comparison to those fed the commercial diet. Furthermore, a lower FCR was observed in birds fed Hy-D-supplemented diets in comparison to those fed commercial diets in the 15 to 28 doa period (Table 4). In the 0 to 40 doa period, BWG, ADG, FI, and ADFI were significantly higher in birds in the Hy-D supplemental dietary treatment in comparison to those in the unsupplemented commercial dietary treatment. Moreover, the treatments that contained 25OHD_3_ had a significantly higher BWG and ADG and a lower FCR than those in the non-injected control treatment, while those in the MDV-alone treatment being intermediate (Table 4).

### 3.3. Meat Yield and Woody Breast Myopathy Score

The processing yield variables of the individually sampled birds that were determined at 14 and 40 doa are shown in Table 5, and those that were determined at 41 doa are shown in Table 6. No significant *in ovo* × dietary treatment interactions were observed for any of the processing variables shown in Table 5 and Table 6. However, BW and P. major and breast meat relative weights at 14 and 40 doa (Table 5), and all the processing variables at 41 doa (Table 6), were significantly higher in birds fed the Hy-D-supplemented diet in comparison to those fed the unsupplemented commercial diet. The pectoralis minor relative weight at 40 doa (Table 5) was also higher in birds in the Hy-D dietary treatment in comparison to those in the unsupplemented commercial dietary treatment.

At 14 doa, the MDV + 25OHD_3_-2.4 *in ovo* treatment led to a higher relative P. minor weight in comparison to that of the birds in the non-injected treatment, with that of the birds in the MDV-alone and MDV + 25OHD_3_-1.2 treatments being intermediate. At 40 doa, a higher P. major relative weight was observed in the MDV + 25OHD_3_-1.2 treatment in comparison to that in both control treatments, with that in the MDV + 25OHD_3_-2.4 *in ovo* treatment being intermediate. However, breast meat yield was higher in the MDV + 25OHD_3_-1.2 *in ovo* treatment in comparison to that in the MDV-alone and non-injected treatments and was higher in the MDV + 25OHD_3_-2.4 treatment when compared with that in the non-injected control treatment, with that in the MDV-alone treatment being intermediate (Table 5).

At 41 doa, P. major and breast meat relative weights were great in any *in ovo* injection treatments containing 25OHD_3_ in comparison to both control treatment groups. The fat pad relative weight was higher in the MDV + 25OHD_3_-1.2 treatment compared to that in the non-injected and MDV-alone treatments, with that in the MDV + 25OHD_3_-2.4 treatment being intermediate. The pectoralis minor relative weight at 41 doa in the MDV + 25OHD_3_-1.2 treatment was higher than that in the non-injected and MDV-alone treatments, and that in the MDV + 25OHD_3_-2.4 treatment was higher than that in the non-injected treatment, with that in the MDV-alone treatment being intermediate (Table 6). There were no significant main or interactive effects on the overall, 0–3, 0 and 1, and 2 and 3 percentage WBM scores in response to either dietary or *in ovo* treatment. However, there was a noticeable trend towards a normal WBM breast score in the Hy-D dietary treatments. The WBM score of 0, indicating a normal breast, tended to be lower (*p* = 0.079) in birds fed commercial diets as compared to those fed a Hy-D supplemental diet (Table 7).

## 4. Discussion

The aim in the current study was to examine the effects of various *in ovo* and dietary 25OHD_3_ levels on the hatching process, live performance, breast meat yield, and incidence of WBM of broilers that received the MDV and that were reared under commercial conditions. The results indicated that there were no noticeable effects of both *in ovo* 25OHD_3_ doses on the hatching process and hatchling quality of birds that received the MDV. It is well documented that the *in ovo* injection of vitamin D sources including 25OHD_3_ at various doses when administered into the amnion shows promising results on the hatchability [40,46,53,54], hatchling quality [1,40,55], posthatch performance [42,43,44,56,57], bone quality [14,56], muscle development [28,42,43,46], and immunity [18,19,45,58] in broilers. Similarly, compared to non-injected or diluent-injected control groups, the *in ovo* administration of the water-soluble form of 25OHD_3_ at various levels ranging from 0.6 to 5.4 µg has been shown to have minimal effects on the hatching process and hatchling quality of broilers that have not received an *in ovo* injection of the MDV [28,42,43,46,58,59]. Dissimilar to the current study, an increase in hatchling BW at 21.9 doi (526 h of incubation) was observed in response to the *in ovo* administration of 1.2 and 2.4 μg of 25OHD_3_ in combination with the MDV when compared to an MDV-alone-injected control group [48]. However, there were no significant effects on hatchling BW at 21 doi due to an MDV-alone-injected treatment or treatments in which the MDV was administered in combination with other 25OHD_3_ levels [48]. The differences in hatching times that were observed in the present study and in the study by Fatemi et al. [48] may be the basis for the inconsistencies in the hatchling BW results. Similar to the current study, it was reported that a slight increase in embryonic mortality occurred in response to high *in ovo* doses of 25OHD_3_ when it was administrated with [48] and without [41] the MDV. Chen et al. [28] reported that the inflammatory response levels of broiler embryos were rapidly stimulated when high doses of 25OHD_3_ were injected at 12 doi and that they remained high until hatch. In addition, pro-inflammatory cytokine expression was increased in 18 doi MDV *in ovo*-injected broilers when 2.4 rather than 0.6 μg of 25OHD_3_ was administered along with the MDV [48]. Thus, an increase in embryonic mortality could be linked to an increase in an immune reaction in response to higher doses of 25OHD_3_. It is worth mentioning that although LEM was increased due to the *in ovo* injection of the MDV in this study, relatively high HS and HI levels (90.4 and 95.1, respectively) were observed when 2.4 μg of 25OHD_3_ was included. Therefore, these findings indicate that high doses of 25OHD_3_ did not interfere with normal levels of hatchability in MDV *in ovo*-injected embryos.

Our findings showed that the dietary or *in ovo* supplementation of 25OHD_3_ increased breast meat yield at 40 and 41 doa in MDV *in ovo*-injected broilers. In addition, the live performance variables (BW, BWG, ADG) from 28 to 40 and 0 to 40 doa were improved. Dietary supplemental 25OHD_3_ likewise increased breast meat yield at 14, 40, and 41 doa and live performance variables throughout the growing phase. Similarly, it is well documented that the breast muscle size at 42 doa increased and that posthatch performance was promoted in response to the addition of 2760 IU/kg of supplemental 25OHD_3_ in feed in comparison to that of an unsupplemented corn–soy bean basal diet that was used throughout the growout phase of broilers [25,60,61]. Furthermore, BW, BWG, and FCR [42,43,44,46] were improved, and breast meat yield was increased [42,43,46] when 2.4 µg of 25OHD_3_ was *in ovo*-injected at 18 doi into the amnion of broiler embryos. These results indicate that both sources of 25OHD_3_ are potent enough to increase breast meat yield but that the dietary source is more effective than the *in ovo*-injected source. Partial reasons for the aforementioned improvement in breast meat yield and live performance in response to 25OHD_3_, regardless of the sources used in this study, could be linked to an enhancement of intestinal histomorphology. Previous studies have shown promising effects as a result of the use of either the dietary or *in ovo* administration of 25OHD_3_. Dietary 25OHD_3_ at a level of 2760 IU/kg in feed has been shown to increase the villus length (**VL**) and VL-to-crypt-depth (**CD**) ratio (**VCR**) and to decrease the CD in 14 and 28 d-old-broilers [23]. In addition, the *in ovo* injection of 2.4 μg of 25OHD_3_ has been shown to increase VL and VCR as compared to non-injected and diluent-injected control groups [19]. Increased VL or VCR is associated with increased nutrient absorption [62], and a shallower CD is associated with a less frequent epithelial cell turnover, leading to a lower energy requirement in the gut [63]. Thus, an improvement in intestinal morphology can allow for the provision of more nutrients for growth and production. It is well observed that an improvement in intestinal morphology is highly associated with the increased breast meat yield and posthatch performance of broilers when vitamin D sources are administered either dietarily [29] or by *in ovo* injection [41,46]. Although the individual effects of dietary and *in ovo* supplemental 25OHD_3_ have been positively associated with improvements in meat yield and posthatch performance, their joint effects need to be tested when these sources of 25OHD_3_ are also used in conjunction with the MDV.

Another reason for the above-mentioned improvement in meat yield and posthatch performance could be linked to a stimulation of genes that are linked to vitamin D activity and growth. In the chicken, a higher expression of 1α-hydroxylase occurs in the kidney, while a secondary increased expression of genes takes place in the muscle [64]. 1α-hydroxylase converts 25OHD_3_ to the active form of D_3_, 1,25-(OH)_2_-D_3_, which is a strong stimulator of growth [65] and muscle development [26,27,28,59]. In addition, 24-hydroxylase expression that converts 25OHD_3_ to the inactive from of vitamin D in the muscle tissue occurs at the same level as 1α-hydroxylase [64]. However, an induced increase in the expression of 1α-hydroxylase over that of 24-hydroxlase can result in greater muscle development and muscle yield. Previous studies have shown that an increase in 1α-hydroxylase expression occurs in response to the *in ovo* injection of 2.4 µg of 25OHD_3_ alone at 28 doa [20]. More recently, 24-hydroxylase expression down-regulation and 1α-hydroxylase expression up-regulation has occurred in hatchlings that received an *in ovo* injection of 1.2 or 2.4 µg of 25OHD_3_ in combination with the MDV when compared to those belonging to non-injected and MDV-alone *in ovo*-injected treatment groups [48]. Therefore, an improvement in the expression of genes including 1α-hydroxylase and 24-hydroxylase, which are involved in vitamin D activity, may partially lead to the positive results observed in breast muscle yield and posthatch performance in response to the 25OHD_3_ sources. However, further research is needed to identify the posthatch expression of genes linked to vitamin D_3_ activity in response to the dietary and *in ovo* sources of 25OHD_3_ administered in combination with the MDV.

Thigh muscle yield at processing was observed in this study to increase as a result of the *in ovo* and dietary administration of 25OHD_3_. Although either source has been shown to significantly impact breast meat yield in this study, the partial increment in thigh meat yield might also be linked to an increase in bone quality. Increases in bone breaking strength, bone mineral content, and leg meat yield were observed in 42-d-old Ross 308 broilers provided supplemental 25OHD_3_ at a level of 2760 IU/kg of feed when compared to those fed supplemental D_3_ at the same level of activity [26]. Furthermore, bone quality and 0 to 28 doa BW were higher in broilers that were injected in the amnion at 17.5 doi with 25OHD_3_ in comparison to those in non-injected and sterilized water-injected treatments [45]. Further study is needed to discover the relationship between an increase in leg meat yield and bone quality when both sources of 25OHD_3_ are administered in combination with the MDV.

The increased occurrence of WMB in the P. major muscle of broilers in response to intensive genetic selection for fast growth and high carcass yield [66] is one of the current major concerns in the poultry industry. There are several factors which are involved in this abnormality, which include broiler age [67], sex [68], genetic selection [69], and broiler diet [67,70]. It is well observed that the breast muscle myopathies and BW and processing body and breast meat yield are less genetically correlated (0.132–0.248) [69]. These data indicate that WBM is mainly correlated to environmental and/or management factors. These contribute 90% of the variance in the incidence of WBM in broiler chickens [69]. Although there were no significant observed effects on the overall WBM score and percentage of moderate or severe levels of WBM for either the dietary or *in ovo* treatments, the trend towards a higher normal breast score in those broilers fed commercial diets as compared to those fed Hy-D diets could be linked to a lower BW, carcass weight, and P. major weight in the commercially fed birds. It is worth mentioning that the overall WBM score was relatively low in this study (approximately 0.5) and that an average breast score (combination of scores 1 and 2) of 90% occurred for all treatment groups regardless of dietary or *in ovo* injection treatments.

It is well-documented that dietary or *in ovo* 25OHD_3_ can reduce systemic and local inflammatory reactions during stressful conditions [19,20,21,26,30,31,32]. An increase in the intestinal expression of genes linked to anti-inflammatory reactions has been observed in response to dietary 25OHD_3_ when broilers are subjected to lipopolysaccharide [30] or coccidiosis [31,32] challenges. Furthermore, the expression of genes linked to reduced inflammation in the breast muscle was significantly up-regulated in the P. major of broilers fed supplemental 25OHD_3_ at a level of 2760 IU/kg of feed [28]. Kuttappan et al. [68,70] reported a significant correlation between chronic inflammation in the breast muscle and an increased incidence of WBM in breast fillets. Although the overall percentage of WBM was relatively low across the treatments in the current study, further research is needed to investigate the effects of both *in ovo* and dietary sources of 25OHD_3_ on WBM incidence over a longer duration of time during growout when a greater percentage of breast fillets commonly exhibit WBM.

## 5. Conclusions

Investigations were performed in the current study concerning the posthatch performance, meat yield, and WBM incidence of broilers in response to the *in ovo* administration of MDV alone or in conjunction with various levels of *in ovo* and dietary 25OHD3. It was observed that compared to non-injected or MDV-alone-injected treatment groups, both sources of 25OHD_3_ were effective in increasing the posthatch breast meat yield and late-phase live performance variables of broilers that received an *in ovo* injection of the MDV. However, the dietary source of 25OHD_3_ was a more potent means by which to improve live performance and breast meat yield throughout the rearing period. No significant treatment effects on WBM scores occurred, while breast meat yield was increased, which indicated that there were beneficial results for meat quality. However, the relationship of this variable with WBM scores over longer periods of time during posthatch growth should be determined. It is suggested that both sources of 25OHD_3_ may be used to promote the posthatch production variables of broilers that receive the MDV by *in ovo* injection.

## Figures and Tables

**Table 1 animals-14-01308-t001:** Feed composition and nutrient composition of experimental diets between 0 and 40 d of age (**doa**).

		Commercial Diet	Hy-D Diet ^1^
		Starter (0–14 doa)	
Item			
Ingredient (%)		Pct	Pct
	Yellow corn	53.23	53.23
	Soybean meal	38.23	38.23
	Animal fat	2.60	2.60
	Dicalcium phosphate	2.23	2.23
	Limestone	1.27	1.27
	Salt	0.34	0.34
	Choline chloride 60%	1.00	1.00
	Lysine	0.28	0.28
	DL-Methionine	0.37	0.37
	L-threonine	0.15	0.15
	Premix ^2^	0.25	0.25
	Hy-D	0.00	0.05
	Coccidiostat ^3^	0.05	0.05
	Total	100	100
Calculated nutrients			
	Crude protein	23	23
	Calcium	0.96	0.96
	Available phosphorus	0.48	0.48
	Apparent metabolizable energy (**AME**; Kcal/kg)	3000	3000
	Digestible methionine	0.51	0.51
	Digestible lysine	1.28	1.28
	Digestible threonine	0.86	0.86
	Digestible total sulfur amino acids (**TSAAs**)	0.95	0.95
	Sodium	0.16	0.16
	Choline	0.16	0.16
		Grower (15–28 doa)	
Item			
Ingredient (%)		Pct	Pct
	Yellow corn	57.13	57.13
	Soybean meal	34.80	34.80
	Animal fat	3.50	3.50
	Dicalcium phosphate	2.00	2.00
	Limestone	1.17	1.17
	Salt	0.34	0.34
	Choline chloride 60%	0.10	0.10
	Lysine	0.21	0.21
	DL-Methionine	0.32	0.32
	L-threonine	0.16	0.16
	Premix	0.25	0.25
	Hy-D	0.00	0.05
	Coccidiostat	0.05	0.05
	Total	100	100
Calculated nutrients			
	Crude protein	21.5	21.5
	Calcium	0.87	0.87
	Available phosphorus	0.435	0.435
	AME (Kcal/kg)	3100	3100
	Digestible methionine	0.47	0.47
	Digestible lysine	1.15	1.15
	Digestible threonine	0.77	0.77
	Digestible TSAA	0.87	0.87
	Sodium	0.16	0.16
	Choline	0.16	0.16
		Finisher (29–45 doa)	
Item			
Ingredient (%)		Pct	Pct
	Yellow corn	54.23	54.23
	Soybean meal	38.23	38.23
	Animal fat	2.50	2.50
	Dicalcium phosphate	2.23	2.23
	Limestone	1.27	1.27
	Salt	0.34	0.34
	Choline chloride 60%	0.10	0.10
	Lysine	0.28	0.28
	DL-Methionine	0.37	0.37
	L-threonine	0.15	0.15
	Premix	0.25	0.25
	Hy-D	0.00	0.05
	Coccidiostat	0.05	0.05
	Total	100	100
Calculated nutrients			
	Crude protein	19.5	19.5
	Calcium	0.78	0.78
	Available phosphorus	0.39	0.39
	AME (Kcal/kg)	3200	3200
	Digestible methionine	0.43	0.43
	Digestible lysine	1.02	1.02
	Digestible threonine	0.68	0.68
	Digestible TSAA	0.80	0.80
	Sodium	0.16	0.16
	Choline	0.16	0.16

^1^ A diet supplemented with 2760 IU/kg feed 25-hydroxyvitamin D_3_. ^2^ The broiler premix provided per kilogram of diet: vitamin A (retinyl acetate), 10,000 IU; cholecalciferol, 250 IU; vitamin E (DL-α-tocopheryl acetate), 50 IU; vitamin K, 4.0 mg; thiamine mononitrate (B_1_), 4.0 mg; riboflavin (B_2_), 10 mg; pyridoxine HCL (B_6_), 5.0 mg; vitamin B_12_ (cobalamin), 0.02 mg; D-pantothenic acid, 15 mg; folic acid, 0.2 mg; niacin, 65 mg; biotin, 1.65 mg; iodine (ethylene diamine dihydroiodide), 1.65 mg; Mn (MnSO_4_H_2_O), 120 mg; Cu, 20 mg; Zn, 100 mg, Se, 0.3 mg; Fe (FeSO_4_.7H_2_O), 800 mg. ^3^ Decocx ^®^ (Zoetis, Parsippany, NJ, USA).

**Table 2 animals-14-01308-t002:** Analyzed dietary values and calculated values of vitamin D_3_ (**D_3_**) and 25-hydroxycholecalciferol (**25OHD_3_**) in the diet.

		D_3_ Calculated	D_3_ Actual	25OHD_3_ Calculated	25OHD_3_ Actual
		-------------------------------IU/kg------------------------------			
Starter					
	Commercial ^1^	250	306	0	ND ^3^
	Hy-D ^2^	250	250	2760	2800
Grower					
	Commercial ^1^	250	191	0	ND
	Hy-D ^2^	250	222	2760	2460
Finisher					
	Commercial ^1^	250	212	0	ND
	Hy-D ^2^	250	274	2760	2350

^1^ D_3_ formulated at 250 IU/kg feed. ^2^ 25-hydroxycholecalciferol formulated at 2760 IU/kg feed and 250 IU/kg feed of D_3_. ^3^ Not detected; the detection limit was 2 μg/kg (equivalent to 80 IU/kg).

**Table 3 animals-14-01308-t003:** Effects of non-injected; and *in ovo* injection treatments of Marek’s disease vaccine (**MDV**) alone or MDV containing various doses of 25-hydroxyvitamin D_3_ (**25OHD_3_**) on mean hatch variables from 0 to 18 d of incubation (**doi**).

Treatment		Egg Weight	PEWL ^1^ 0–12	PEWL ^1^ 12–18	PEWL ^1^ 0–18	HS ^1^	HI ^1^	LEM ^2^	PPM ^3^	PEM ^4^	Hatchling Mortality ^5^	Hatchling BW
	n	---g---	---------------------------------------------%------------------------------------------------									---g---
Non-injected ^6^	16	55.5	3.83	3.33	7.16	94.1	97.6	1.13 ^b^	0.23	0.93	0.23	44.6
MDV ^7^	16	55.4	3.84	3.94	7.78	91.3	95.5	4.25 ^a^	0.25	0.23	0	43.5
MDV + 25OHD_3_-1.2 ^7,8^	16	55.3	3.87	3.55	7.42	93.3	95.8	3.25 ^ab^	0.23	0.47	0.46	44.6
MDV + 25OHD_3_-2.4 ^7,9^	16	55.1	3.88	3.66	7.54	90.4	95.1	4.38 ^a^	0	0.99	0	44.2
SEM		0.13	0.057	0.289	0.308	1.50	1.42	1.135	0.287	0.555	0.263	0.74
*p*-value		0.090	0.752	0.244	0.279	0.082	0.352	**0.042**	0.801	0.480	0.275	0.453

^a-b^ Treatment means within the same variable column lacking a common superscript differ significantly (*p*≤ 0.05). ^1^ Percentage egg weight loss (**PEWL**) between 0 and 12, 12 and 18, and 0 and 18 doi; hatchability of set eggs (**HS**); hatchability of injected live embryonated eggs (**HI**), hatchling body weight (**BW**). ^2^ Late embryo mortality (between 18 and 21 doi prior to the piping process). ^3^ Embryo mortality between 18 and 21 doi during the pipping process. ^4^ Mortality after hatchlings immediately complete shell emergence and prior to their placement at 21 doi. ^5^ Mortality of hatchlings at placement at 21 doi. ^6^ Embryos that did not receive a solution injection. ^7^ Received a 50 µL solution volume of the Marek’s disease vaccine injected at 18 doi. ^8^ Embryos injected with the Marek’s disease vaccine containing 1.2 μg of 25OHD_3_. ^9^ Embryos injected with the Marek’s disease vaccine containing 2.4 μg of 25OHD_3_.

**Table 4 animals-14-01308-t004:** Effects of non-injected and *in ovo* injection treatments of Marek’s disease vaccine (**MDV**) alone or MDV containing various doses of 25-hydroxyvitamin D_3_ (**25OHD_3_**), and commercial diets or diets supplemented with 2760 IU/kg of 25OHD_3_ on mean live performance variables throughout 40 d of age (**doa**).

Treatment		BW ^1^ (g)	BWG ^1^ (g)	ADG ^1^ (g)	FI ^1^ (g)	ADFI ^1^ (g)	FCR ^1^ (g/g)
		-------------------------Starter (0 to14 doa)-----------------------					
*In ovo* injection							
	Non-injected ^2^	463	418	29.9	508	36.3	1.22
	MDV ^3^	456	412	29.4	509	36.4	1.24
	25OHD_3_-1.2 ^4^	447	402	28.7	497	35.5	1.24
	25OHD_3_-2.4 ^5^	458	414	29.6	500	35.7	1.21
	SEM	7.8	7.5	0.55	12.2	1.58	0.032
Diet							
	Commercial	438 ^b^	394 ^b^	28.1 ^b^	490 ^b^	35.0 ^b^	1.25
	Hy-D ^6^	474 ^a^	429 ^a^	30.6 ^a^	517 ^a^	37.0 ^a^	1.21
	SEM	3.9	5.8	0.39	8.6	6.30	0.023
		*p*-value					
*In ovo* Diet *In ovo* × diet		0.198	0.190	0.190	0.723	0.723	0.765
		<0.0001	<0.0001	<0.0001	0.003	0.003	0.092
		0.174	0.156	0.157	0.300	0.300	0.922
		BW (g)	BWG (g)	ADG (g)	FI (g)	ADFI (g)	FCR (g/g)
		------------------Grower (15 to 28 doa)------------------					
*In ovo* injection							
	Non-injected	1439	1020	73	1451	96	1.42
	MDV	1438	1027	73	1433	97	1.39
	25OHD_3_-1.2	1447	1045	75	1452	101	1.39
	25OHD_3_-2.4	1456	1043	75	1460	101	1.41
	SEM	30.0	27.5	2.0	26.5	4.9	0.004
Diet							
	Commercial	1244 ^b^	850 ^b^	61 ^b^	1246 ^b^	89 ^b^	1.47 ^a^
	Hy-D	1646 ^a^	1217 ^a^	87 ^a^	1652 ^a^	107 ^a^	1.34 ^b^
	SEM	21.2	19.4	1.4	18.7	3.5	0.025
		----------------------------*p*-value----------------------------					
*In ovo* Diet *In ovo* × diet		0.923	0.765	0.765	0.780	0.647	0.781
		<0.0001	<0.0001	<0.0001	<0.0001	<0.0001	<0.0001
		0.085	0.152	0.152	0.287	0.966	0.590
		BW (g)	BWG (g)	ADG (g)	FI (g)	ADFI (g)	FCR (g/g)
		---------------------Finisher (29 to 40 doa)------------------					
*In ovo* injection							
	Non-injected	2382 ^b^	944 ^b^	78.7 ^b^	1734	145	1.84
	MDV	2465 ^ab^	1027 ^ab^	85.5 ^ab^	1718	145	1.71
	25OHD_3_-1.2	2557 ^a^	1111 ^a^	92.6 ^a^	1744	143	1.58
	25OHD_3_-2.4	2552 ^a^	1096 ^a^	91.3 ^a^	1740	146	1.60
	SEM	56.0	58.9	4.91	44.2	3.7	0.109
Diet							
	Commercial	2100 ^b^	857 ^b^	71.4 ^b^	1341 ^b^	112 ^b^	1.61
	Hy-D	2878 ^a^	1231 ^a^	102.6 ^a^	2127 ^a^	177 ^a^	1.75
	SEM	39.6	41.7	3.47	31.3	1.9	0.077
				*p*-value			
*In ovo* Diet *In ovo* × diet		0.010	0.030	0.031	0.942	0.944	0.095
		<0.0001	<0.0001	<0.0001	<0.0001	<0.0001	0.079
		0.810	0.899	0.900	0.198	0.197	0.657
		BWG (g)	ADG (g)	FI (g)	ADFI (g)	FCR (g/g)	Total Mortality (%)
		--------------------------(0 to 40 doa)-----------------------------					
*In ovo* injection							
	Non-injected	2337 ^b^	58.4 ^b^	3694	92.3	1.58 ^a^	3.21
	MDV	2424 ^ab^	60.6 ^ab^	3660	91.5	1.51 ^ab^	3.85
	25OHD_3_-1.2	2516 ^a^	62.9 ^a^	3692	92.3	1.47 ^b^	0.64
	25OHD_3_-2.4	2511 ^a^	62.8 ^a^	3700	92.5	1.48 ^b^	3.85
	SEM	56.6	1.41	64.1	1.61	0.042	1.789
Diet							
	Commercial	2062 ^b^	51.5 ^b^	3077 ^b^	76.9 ^b^	1.50	3.53
	Hy-D	2833 ^a^	70.8 ^a^	4297 ^a^	107.4 ^a^	1.52	2.25
	SEM	40.0	1.00	32.1	1.14	0.029	1.265
		----------------------------*p*-value----------------------------					
*In ovo*		0.009	0.009	0.924	0.925	0.042	0.243
Diet		<0.0001	<0.0001	<0.0001	<0.0001	0.525	0.317
*In ovo* × diet		0.783	0.785	0.113	0.8159	0.832	0.915

^a-b^ Treatment means within the same variable column lacking a common superscript differ significantly (*p* ≤ 0.05). ^1^ BW, BW gain (**BWG**), average daily gain (**ADG**), feed intake (**FI**), average daily feed intake (**ADFI**), feed conversion ratio (**FCR**), and total mortality. ^2^ Embryos that did not receive a solution injection. ^3^ Received a 50 µL solution volume of the Marek’s disease vaccine injected at 18 doi. ^4^ Embryos injected with the Marek’s disease vaccine containing 1.2 μg of 25OHD_3_. ^5^ Embryos injected with the Marek’s disease vaccine containing 2.4 μg of 25OHD_3_. ^6^ A diet supplemented with 2650 IU/kg 25OHD_3_ throughout the rearing period.

**Table 5 animals-14-01308-t005:** Effects of non-injected and *in ovo* injection treatments of Marek’s disease vaccine (**MDV**) alone or MDV containing various doses of 25-hydroxyvitamin D_3_ (**25OHD_3_**), and commercial diets or diets supplemented with 2760 IU/kg of 25OHD_3_ on mean sample BW and relative weights of pectoralis major (**P. major**) and minor (**P. minor**) to BW at 14 and 40 d of age (**doa**).

Treatment		BW (g)	P. Major (%)	P. Minor (%)	Breast (%)
		14 Doa			
*In ovo* injection					
	Non-injected ^1^	453	13.83	2.56 ^b^	16.39
	MDV ^2^	479	14.67	2.68 ^ab^	17.35
	MDV + 25OHD_3_-1.2 ^3^	750	14.74	2.75 ^ab^	17.49
	MDV + 25OHD_3_-2.4 ^4^	466	14.52	2.88 ^a^	17.40
	SEM	18.5	0.528	0.102	0.543
Diet					
	Commercial	445 ^b^	13.92 ^b^	2.68	16.60 ^b^
	Hy-D ^5^	479 ^a^	14.96 ^a^	2.76	17.72 ^a^
	SEM	13.1	0.373	0.080	0.384
		*p*-value			
*In ovo*		0.397	0.307	0.050	0.164
Diet		0.014	0.009	0.310	0.006
*In ovo* × diet		0.298	0.941	0.425	0.954
		40 doa			
*In ovo* injection					
	Non-injected	2522	17.4 ^b^	3.56	20.9 ^c^
	MDV	2720	18.0 ^b^	3.44	21.4 ^bc^
	MDV + 25OHD_3_-1.2	2786	20.0 ^a^	3.62	23.5 ^a^
	MDV + 25OHD_3_-2.4	2729	19.1 ^ab^	3.78	22.9 ^ab^
	SEM	158.6	0.90	0.161	0.93
Diet					
	Commercial	2323 ^b^	16.3 ^b^	3.38 ^b^	23.8 ^b^
	Hy-D	3005 ^a^	21.0 ^a^	3.82 ^a^	25.7 ^a^
	SEM	112.2	0.63	0.114	0.72
		----------------------------*p*-value----------------------------			
*In ovo*		0.534	0.029	0.211	0.023
Diet		<0.0001	<0.0001	0.001	<0.0001
*In ovo* × diet		0.839	0.210	0.817	0.284

^a-c^ Treatment means within the same variable column lacking a common superscript differ significantly (*p* ≤ 0.05). ^1^ Embryos that did not receive a solution injection. ^2^ Received a 50 µL solution volume of the Marek’s disease vaccine injected at 18 doi. ^3^ Embryos injected with the Marek’s disease vaccine containing 1.2 μg of 25OHD_3_. ^4^ Embryos injected with the Marek’s disease vaccine containing 2.4 μg of 25OHD_3_. ^5^ A diet supplemented with 2650 IU/kg 25OHD_3_ throughout the rearing period.

**Table 6 animals-14-01308-t006:** Effects of non-injected and *in ovo* injection treatments of Marek’s disease vaccine (**MDV**) alone or MDV containing various doses of 25-hydroxyvitamin D_3_ (**25OHD_3_**), and commercial diets or diets supplemented with 2760 IU/kg of 25OHD_3_ on mean processing parts at 41 d of age (**doa**).

Treatment		Carcass	Fat Pad	Wings	Drumsticks	Thighs	P. Major	P. Minor	Breast
		(g)	---------------------------------------(%)-----------------------------------------						
*In ovo* injection									
	Non-injected ^1^	2132	0.110 ^b^	9.31	11.5	13.9 ^b^	25.7 ^b^	5.09 ^c^	30.7 ^b^
	MDV ^2^	2176	0105 ^b^	9.47	11.5	14.4 ^ab^	26.5 ^b^	5.10 ^bc^	31.6 ^b^
	MDV + 25OHD_3_-1.2 ^3^	2222	0.121 ^a^	9.61	11.9	14.7 ^a^	27.7 ^a^	5.36 ^a^	33.0 ^a^
	MDV + 25OHD_3_-2.4 ^4^	2210	0.111 ^ab^	9.55	11.7	14.7 ^a^	27.8 ^a^	5.29 ^ab^	33.1 ^a^
	SEM	43.6	0.0051	0.144	0.19	0.30	0.52	0.096	0.58
Diet									
	Commercial	2075 ^b^	0.102 ^b^	9.20 ^b^	11.2 ^b^	13.6 ^b^	25.6 ^b^	5.04 ^b^	30.6 ^b^
	Hy-D ^5^	2295 ^a^	0.122 ^a^	9.77 ^a^	12.0 ^a^	15.2 ^a^	28.2 ^a^	5.38 ^a^	33.6 ^a^
	SEM	30.9	5.8	0.102	0.13	0.21	0.37	0.068	0.41
		--------------------------------------------------------*p*-value---------------------------------------------------							
*In ovo*		0.181	0.030	0.200	0.147	0.033	0.001	0.014	0.001
Diet		<0.0001	<0.0001	<0.0001	<0.0001	<0.0001	<0.0001	<0.0001	<0.0001
*In ovo* × diet		0.938	0.445	0.611	0.554	0.907	0.672	0.331	0.738

^a-c^ Treatment means within the same variable column lacking a common superscript differ significantly (*p* ≤ 0.05). ^1^ Embryos that did not receive a solution injection. ^2^ Received a 50 µL solution volume of the Marek’s disease vaccine injected at 18 doi. ^3^ Embryos injected with the Marek’s disease vaccine containing 1.2 μg of 25OHD_3_. ^4^ Embryos injected with the Marek’s disease vaccine containing 2.4 μg of 25OHD_3_. ^5^ A diet supplemented with 2650 IU/kg 25OHD_3_ throughout the rearing period.

**Table 7 animals-14-01308-t007:** Effects of non-injected and *in ovo* injection treatments of Marek’s disease vaccine (**MDV**) alone or MDV containing various doses of 25-hydroxyvitamin D_3_ (**25OHD_3_**), and commercial diets or diets supplemented with 2760 IU/kg of 25OHD_3_ on incidence of woody breast myopathy scores at 41 d of age (**doa**).

Treatment		Overall	Score 0	Score 1	Score 2	Score 3	Score 0 and 1	Score 2 and 3
			(%)					
*In ovo* injection								
	Non-injected ^1^	0.55	61.8	29.3	4.5	4.5	91.1	8.9
	MDV ^2^	0.42	70.5	21.9	4.3	3.3	92.4	7.6
	MDV + 25OHD_3_-1.2 ^3^	0.46	71.8	16.9	7.5	3.9	88.7	11.3
	MDV + 25OHD_3_-2.4 ^4^	0.43	64.5	27.5	6.1	2.00	92.0	8.0
	SEM	0.111	64.41	5.42	3.08	2.16	3.47	3.47
Diet								
	Commercial	0.43	71.1	21.2	5.3	2.4	92.3	7.8
	Hy-D ^5^	0.51	63.2	26.6	5.9	4.4	89.7	10.2
	SEM	0.078	4.51	3.83	2.18	1.53	2.46	2.56
		----------------------------------*p*-value----------------------------------						
*In ovo*		0.616	0.328	0.107	0.710	0.685	0.707	0.707
Diet		0.312	0.079	0.170	0.793	0.194	0.298	0.298
*In ovo* × diet		0.772	0.438	0.164	0.955	0.180	0.565	0.565

^1^ Embryos that did not receive a solution injection. ^2^ Received a 50 µL solution volume of the Marek’s disease vaccine injected at 18 doi. ^3^ Embryos injected with the Marek’s disease vaccine containing 1.2 μg of 25OHD_3_. ^4^ Embryos injected with the Marek’s disease vaccine containing 2.4 μg of 25OHD_3_. ^5^ A diet supplemented with 2650 IU/kg 25OHD_3_ throughout the rearing period.

## Data Availability

No data from this study were deposited in an official repository.

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
