# Peer review of "Effects of the Marek’s Disease Vaccine on the Performance, Meat Yield, and Incidence of Woody Breast Myopathy in Ross 708 Broilers When Administered Alone or in Conjunction with In ovo and Dietary Supplemental 25-Hydroxycholecalciferol"

_animals, 2024, doi:10.3390/ani14091308_

Round 1
Reviewer 1 Report
Comments and Suggestions for Authors
“Effects of the in ovo administration of the Marek's Disease Vaccine alone or in combination with the supplemental in ovo and dietary administration of 25-hydroxyvitamin D3 on the performance, meat yield, and incidence of woody breast myopathy in Ross 708 broilers”
The study investigated the effects of different levels of in-ovo and dietary supplementation of 25-hydroxyvitamin D3 (25OHD3) on various parameters related to broiler production and meat quality. The study explicitly focuses on the impact on hatch variables, live performance, breast meat yield, and the incidence of woody breast myopathy (WBM). Additionally, the study aimed to evaluate the potential synergistic effects of 25OHD3 supplementation with the Marek's Disease Vaccine (MDV) in broilers.
Abstract: The abstract is well-designed and provides a clear outline of the study by stating the study's objective, the methodologies used for testing the hypothesis, and the results found on the investigated aspects. The results were briefly summarized and presented, acknowledging the need for further research to determine the physiological mechanisms promoting the findings.
Introduction: The introduction clearly states the objectives of the study, and its content aligns with the abstract, results, discussion, and conclusion sections of the manuscript. This section provides a comprehensive background on the use of in-ovo injections and the importance of vitamin D in broiler production. Moreover, it summarizes previous research on in-ovo injection of 25OHD3 and its effects on hatchability, post-hatch performance, and meat yield, aligning with the findings in the results and discussion sections of the manuscript. This section also considers the immune responses in the presence of 25OHD3, although, in a very brief way.
Materials and methods: The experimental design and statistical analysis seem to be well done, and there are no comments from the reviewer.
Results: Overall, the results indicate that in-ovo and dietary supplementation of 25OHD3 in combination with the MDV, have significant effects on broiler performance and meat yield, with potential implications for poultry production. The results section seems to be well written, and the findings are well explained. There are no comments from the reviewer.
Discussion: The discussion section interprets the results presented in the study, providing insights into the observed effects of in-ovo injection of 25OHD3 alone and in combination with MDV on broiler performance, meat yield, and quality, which aligns with the abstract, introduction, results, and conclusion sections of the manuscript. Importantly, discussion section is consistent with previous literature. This section contrasts and correlates specific data and findings from the results section with data from the available literature to contextualize the study's findings and to support the manuscript's conclusions. Furthermore, the discussion section clearly addresses the objectives outlined in the introduction. Thus, it shows that the outcomes align with the study goals, highlighting the implications of the findings and suggesting paths for future research.
Conclusion: the conclusion section summarizes the essential findings of the study, and emphasize the effectiveness of in-ovo and dietary sources of 25OHD3 in increasing post-hatch breast meat yield and improving late-phase live performance variables. The effect of the two sources of 25OHD3 is highlighted, and it was stated that both treatments were beneficial. However, it shows that the vitamin D dietary source demonstrated greater potency in improving live performance and breast meat yield compared to the In-ovo injection. The conclusion section also highlights the absence of significant treatment effects on WBM scores.Moreover, this section acknowledges the need for further research.
Major comments:
Dear authors.
There is not much information through the whole manuscript about meat quality. It is vital for this manuscript to define meat quality in the document and to discuss exactly how the treatments and the findings affect meat quality. Please correct.
Author Response
Reviewer 1
The study investigated the effects of different levels of in-ovo and dietary supplementation of 25-hydroxyvitamin D3 (25OHD3) on various parameters related to broiler production and meat quality. The study explicitly focuses on the impact on hatch variables, live performance, breast meat yield, and the incidence of woody breast myopathy (WBM). Additionally, the study aimed to evaluate the potential synergistic effects of 25OHD3 supplementation with the Marek's Disease Vaccine (MDV) in broilers.
Abstract: The abstract is well-designed and provides a clear outline of the study by stating the study's objective, the methodologies used for testing the hypothesis, and the results found on the investigated aspects. The results were briefly summarized and presented, acknowledging the need for further research to determine the physiological mechanisms promoting the findings.
Answer:
Thank you for the comment
Introduction: The introduction clearly states the objectives of the study, and its content aligns with the abstract, results, discussion, and conclusion sections of the manuscript. This section provides a comprehensive background on the use of in-ovo injections and the importance of vitamin D in broiler production. Moreover, it summarizes previous research on in-ovo injection of 25OHD3 and its effects on hatchability, post-hatch performance, and meat yield, aligning with the findings in the results and discussion sections of the manuscript. This section also considers the immune responses in the presence of 25OHD3, although, in a very brief way.
Answer:
Thank you for the comment
Materials and methods: The experimental design and statistical analysis seem to be well done, and there are no comments from the reviewer.
Answer:
Thank you for the comment
Results: Overall, the results indicate that in-ovo and dietary supplementation of 25OHD3 in combination with the MDV, have significant effects on broiler performance and meat yield, with potential implications for poultry production. The results section seems to be well written, and the findings are well explained. There are no comments from the reviewer.
Answer:
Thank you for the comment
Discussion: The discussion section interprets the results presented in the study, providing insights into the observed effects of in-ovo injection of 25OHD3 alone and in combination with MDV on broiler performance, meat yield, and quality, which aligns with the abstract, introduction, results, and conclusion sections of the manuscript. Importantly, discussion section is consistent with previous literature. This section contrasts and correlates specific data and findings from the results section with data from the available literature to contextualize the study's findings and to support the manuscript's conclusions. Furthermore, the discussion section clearly addresses the objectives outlined in the introduction. Thus, it shows that the outcomes align with the study goals, highlighting the implications of the findings and suggesting paths for future research.
Answer:
Thank you for the comment
Conclusion: the conclusion section summarizes the essential findings of the study, and emphasize the effectiveness of in-ovo and dietary sources of 25OHD3 in increasing post-hatch breast meat yield and improving late-phase live performance variables. The effect of the two sources of 25OHD3 is highlighted, and it was stated that both treatments were beneficial. However, it shows that the vitamin D dietary source demonstrated greater potency in improving live performance and breast meat yield compared to the In-ovo injection. The conclusion section also highlights the absence of significant treatment effects on WBM scores. Moreover, this section acknowledges the need for further research.
Answer:
Thank you for the comment
Major comments:
Dear authors.
There is not much information through the whole manuscript about meat quality. It is vital for this manuscript to define meat quality in the document and to discuss exactly how the treatments and the findings affect meat quality. Please correct.
Answer:
Thank you for the suggestion, the relevant corrections were applied to the manuscript in both Introduction and Discussion section. In the introduction the corrections were made on lines 96-108 and in the Discussion on lines 426 – 444.
Reviewer 2 Report
Comments and Suggestions for Authors
1. The title showing the research studied “the performance, meat yield, and incidence of woody breast myopathy in Ross 708 broilers.” However, in the Introduction, on any information of the woody breast myopathy.
2. Usually, the problem of the broilers industry should be showed in the beginning of Introduction. However, no any problems of broilers industry be showed, So, what kinds of problem should be solved is unknown. The Introduction is short of logic.
3. This study used 35 week-old commercial Ross 708 broiler breeder hens, In Table 3. The eggs weight is 82.6-83.2g,usually,50-60g is actually, the data should be double checked.
4. In Table 3. hatchability of set eggs(HS), usually, the hatchability should be calculated based on the fertile egg, not set eggs. Hatchling Mortality,LEM,PPM, PEM should based on fertile egg also. the replicate is only 3 in the hatch experiment,the result is unbelievable.
5.In the discussion. 424-429 The intestinal expression of genes linked to anti-inflammatory responses has been observed in conjunction with their up-regulation after a lipopolysaccharide [32] or coccidiosis [33,34] challenge. Furthermore, the expression of genes linked to reduced inflammation in the breast muscle was also observed to occur in the P. major of broilers fed supplemental 25OHD3 at a level of 2,760 IU/kg of feed [28]. However, no any anti-infalmmatory items were studied. The discussion can’t explain anything.
6. The aim in the current study was to examine the effects of various in ovo and dietary25OHD3 levels on the hatching process, live performance, breast meat yield, and incidence of WBM of broilers that received the MDV and that were reared under commercial conditions. The results is different between in ovo and dietary supplemental 25OHD. Usually, the effect of ovo can be extend to the young broiler, however, there were no any interaction effects occurred for any items in this research among the dietary or in ovo injection treatments. It is hard to believe.
6. In the abstract, the conclusion is not accurate and completely.
Author Response
- The title showing the research studied “the performance, meat yield, and incidence of woody breast myopathy in Ross 708 broilers.” However, in the Introduction, on any information of the woody breast myopathy.
Answer:
The relevant changes were applied to the introduction. Lines 93-108
“However, this rapid growth rate has been shown to be linked to increased metabolic diseases and abnormalities in breast fillets exhibiting woody breast myopathy (WBM) as a result of an increase in myodegeneration, lipidosis, fibrosis, and oxidative stress, and a decrease in protein synthesis in pectoralis (P.) major [37]. It is estimated that some level of WMB can be detected in approximately 9% of the breast fillets of larger broilers (2.72–4.53 kg) [38]. It is worth-mentioning that WMB not only negatively affects meat quality, but also has negative effects on production profits. This is largely due to the fact that severe WBM breasts are not allowed to be sold and that breasts with moderate WBM scores are usually sold at half-price [37,39]. It suggested that nutritional agents that can increase protein synthesis and reduce inflammation in fillets can be used to reduce WBM incidence [37,39]. Therefore, vitamin D3 sources that stimulate immunocompetent and muscle synthesis may lower WBM incidence.”
- Usually, the problem of the broilers industry should be showed in the beginning of Introduction. However, no any problems of broilers industry be showed, So, what kinds of problem should be solved is unknown. The Introduction is short of logic.
Answer:
The relevant information were added to introduction. Lines 56-71.
“It is well-observed that the important and pragmatic innovation for the past 3 decades that has been shown a significant influence on the poultry industry is in ovo injection technology. It has been designed to provide a stress-free, earlier, faster, and uniform delivery of vaccines for the protection of broilers and it has been emerged as an alternative approach to the posthatch vaccination of chickens, particularly in broilers [1-4]. In ovo injection is used for the direct administration of particular nutrients or vaccines in the amnion of embryos between 17.5 and 19.25 d of incubation (doi) [2-4]. Furthermore, the 18 doi in ovo administration of several vaccines for Escherichia Coli [5], Mycoplasma galicepticum [6], infectious bursa disease [7] and Marek's disease [8], and various supplemental nutrients including vitamins, minerals, and carbohydrates [1,2,9,10,11] have been shown to promote not only hatchling immunity, but also to improve hatchability and posthatch performance. Although the use of various vaccine has been well-established in the US broiler`s hatcheries, the use of any nutrients including vitamins minerals, proteins, amino acids, and organic acids has not been commercially used. One major reason could be due to lack of their combined effects with any commercial vaccines and their subsequent effects on the broiler production variables have not been investigated”.
- This study used 35 week-old commercial Ross 708 broiler breeder hens, In Table 3. The eggs weight is 82.6-83.2g,usually,50-60g is actually, the data should be double checked.
Answer:
Thanks for the comment the relevant corrections were applied to Table 3.
- In Table 3. hatchability of set eggs(HS), usually, the hatchability should be calculated based on the fertile egg, not set eggs. Hatchling Mortality, LEM, PPM, PEM should based on fertile egg also. the replicate is only 3 in the hatch experiment, the result is unbelievable.
Answer:
In this study, not only the hatchability of set eggs (HS) presented in Table 3, but also the hatchability of live embryonated eggs (HI) which is represented as live embryo and fertile eggs showed in Table 3. All hatch residue analysis including Hatchling Mortality, LEM, PPM, PEM are calculated based on those eggs that passed candling at 18 doi. The relevant information were inserted into the text in order to clear this statement. Lines 163-166
“Hatch residue analysis was performed on eggs after candling between 18 to 21 doi, according to the procedure described by Fatemi et al. [48], to ensure that only live embrocated eggs were included until hatch. This analysis included late embryonic mortality (LEM), prior piped embryonic mortality (PPM), post piped embryonic mortality (PEM), and hatchling mortality.”
The number of replication per treatment is not 3 and it is 16 and the relevant correction was added to the table 3.
5.In the discussion. 424-429 The intestinal expression of genes linked to anti-inflammatory responses has been observed in conjunction with their up-regulation after a lipopolysaccharide [32] or coccidiosis [33,34] challenge. Furthermore, the expression of genes linked to reduced inflammation in the breast muscle was also observed to occur in the P. major of broilers fed supplemental 25OHD3 at a level of 2,760 IU/kg of feed [28]. However, no any anti-infalmmatory items were studied. The discussion can’t explain anything.
Answer:
The inflammatory indictors were reported in companion previous study where the similar treatments used in this study were used in previous study conducted by Fatemi et al., 2024 with the exact same procedure. Therefore, partial of this experiment was replicated of pervious study that allowed us to refer to our pervious findings. We also tested the effects of 25OHD3 in ovo injection on the performance and immune-related gene expression of broilers subjected to coccidiosis challenge and found that the same treatments that were tested in this study with the same method of perpetration resulted in changes in inflammatory response genes as well as improvement in the performance variable during severe infection; this we speculate form previous finding and proposed the possible reason for the results of current study. Also, we also indicated that this hypothesis needs to be tested for the current study in order to be conservative enough about our speculation.
- The aim in the current study was to examine the effects of various in ovo and dietary25OHD3 levels on the hatching process, live performance, breast meat yield, and incidence of WBM of broilers that received the MDV and that were reared under commercial conditions. The results is different between in ovo and dietary supplemental 25OHD. Usually, the effect of ovo can be extend to the young broiler, however, there were no any interaction effects occurred for any items in this research among the dietary or in ovo injection treatments. It is hard to believe.
Answer:
Thanks for the comment, however, we design the experiment to compare the effect of in ovo and dietary 25OHD3 although they performed individually better than control group, it sounds that the dietary is more potent, as we mentioned in the discussion, than in ovo. Thus, in order to see possible interaction effects on performance and meat yield, it may need to raise vitamin D3 at the same level of 25OHD3 (2,670 IU/kg) to give more chance to in ovo to show its potential. We were unable to do this hypothesis in this trial because we needed to be consistent with our previous research that D3 was used at 250 IU/ kg, otherwise our results were not be comparable. Also, we could include 4 more treatments in order to considered above-mentioned comparison due to insufficient pens for appropriate replication per treatment.
- In the abstract, the conclusion is not accurate and completely.
Answer:
The relevant changes were applied in to the Abstract.
“No significant main or interaction effects occurred for WBM among the dietary or in ovo injection treatments. However, in response to in ovo 25OHD3 supplementation, BW and BWG in the 29 to 40 doa period, and BWG and FCR in the 0 to 40 doa period, improved. In addition, at 40 and 41 doa, breast meat yield increased in response to in ovo and dietary 25OHD3 supplementation. Future research is needed to determine the possible reasons that may have been involved in the aforementioned improvements.”
Reviewer 3 Report
Comments and Suggestions for Authors
It is my utmost pleasure to review this high-quality research and well-written manuscript investigating the effect of 25-hydroxyvitamin D3 supplementation (in ovo, dietary, or both) combined with in ovo administration of the Marek's Disease vaccine on broilers` hatching parameters, growth performance, carcass and parts yield, and woody breast myopathy. However, I believe that the manuscript needs a few minor refinements.
Line (L) 44: “At 14 and 40, 41 d of….” I would change “At 14, 40, and 41 d of….”.
L 91: “in in comparison...” remove the extra “in”.
L 146: “broilers were placed at a 062 m2/bird stocking density…..” I think that there is a missing decimal point.
L 142: Where the birds were processed?
L 227: “that in in the non-injected…” remove the extra “in”.
L 315: “…..dietary25OHD3 levels….” missing space before “25OHD3”.
L 424- 425: “The intestinal expression of genes linked to anti-inflammatory responses has been observed in conjunction with their up-regulation after a lipopolysaccharide…..”
Do you mean that upregulation of genes linked to anti-inflammatory responses was observed after a lipopolysaccharide and coccidiosis challenge?
Please rewrite to reduce the confusion.
L 427- 429: “..the expression of genes linked to reduced inflammation in the breast 427 muscle was also observed to occur in the P. major of broilers fed supplemental 25OHD3 at a level of 2,760 IU/kg of feed”
How was the change in the expression level, was it significantly elevated? Or you meant that the expression level of genes linked to reduced inflammation was observed while normally it should not be observed? Please reform the sentence accordingly.
Author Response
It is my utmost pleasure to review this high-quality research and well-written manuscript investigating the effect of 25-hydroxyvitamin D3 supplementation (in ovo, dietary, or both) combined with in ovo administration of the Marek's Disease vaccine on broilers` hatching parameters, growth performance, carcass and parts yield, and woody breast myopathy. However, I believe that the manuscript needs a few minor refinements.
Answer:
Thank you for the review and comments
- Line (L) 44: “At 14 and 40, 41 d of….” I would change “At 14, 40, and 41 d of….”.
Answer:
The relevant change was applied in the abstract
- L 91: “in in comparison...” remove the extra “in”.
Answer:
The relevant change was applied on line 113
- L 146: “broilers were placed at a 062 m2/bird stocking density…..” I think that there is a missing decimal point.
Answer:
The relevant change was applied on line 169
- L 142: Where the birds were processed?
Answer:
The relevant change was applied on line 177-178
“Birds were processed at 41 doa at Mississippi State University poultry farm processing plant.”
- L 227: “that in in the non-injected…” remove the extra “in”.
Answer:
The relevant change was applied on line 224
- L 315: “…..dietary25OHD3 levels….” missing space before “25OHD3”.
Answer:
The relevant change was applied on line 328
- L 424- 425: “The intestinal expression of genes linked to anti-inflammatory responses has been observed in conjunction with their up-regulation after a lipopolysaccharide…..”
Do you mean that upregulation of genes linked to anti-inflammatory responses was observed after a lipopolysaccharide and coccidiosis challenge?
Please rewrite to reduce the confusion.
Answer:
The relevant change was applied on line 443-444
“An increase in the intestinal expression of genes linked to anti-inflammatory reaction has been observed in response to dietary 25OHD¬3 when broilers subjected to lipopolysaccharide”
- L 427- 429: “..the expression of genes linked to reduced inflammation in the breast 427 muscle was also observed to occur in the P. major of broilers fed supplemental 25OHD3 at a level of 2,760 IU/kg of feed”
How was the change in the expression level, was it significantly elevated? Or you meant that the expression level of genes linked to reduced inflammation was observed while normally it should not be observed? Please reform the sentence accordingly.
Answer:
Inflammation at any levels in any tissue is not supposed to occur under normal circumstances unless birds are invaded with enteritis pathogens or other stimuli agents cause an increase in inflammation.
The relevant change was applied on line 446-448
“Furthermore, the expression of genes linked to reduced inflammation in the breast muscle was significantly up-regulated in the P. major of broilers fed supplemental 25OHD3 at a level of 2,760 IU/kg of feed”
Round 2
Reviewer 2 Report
Comments and Suggestions for Authors
1.Cu, 20 mg; Zn, 100 mg, Se, 0.3 mg; Fe (FeSO4.7H2O), 800 mg isn't right. Cu, Zn ,Se should supply the sources like Fe.
2.the title of Table 3-6 is too long, should be shorted.
3.Table 1 title should be the "Feed composition and nutrient composition of experimental diet. Table 2 the title should be " Analyzed dietary values and calculated values of vitamin D3 (D3) and 25-hydroxycholecalciferol (25OHD3) in the diet
Author Response
1.Cu, 20 mg; Zn, 100 mg, Se, 0.3 mg; Fe (FeSO4.7H2O), 800 mg isn't right. Cu, Zn ,Se should supply the sources like Fe.
Answer:
Thank you for the comment; however, we just reported the information provided by the product used in this study and there was no additional information with regards to the source of Cu, Zn , and Se in the premixed used. Also, this format has been used in pervious publications for example: Fatemi et al., 2021a-d
2.the title of Table 3-6 is too long, should be shorted.
Answer:
The relevant changes were applied to the corresponding tables
3.Table 1 title should be the "Feed composition and nutrient composition of experimental diet. Table 2 the title should be " Analyzed dietary values and calculated values of vitamin D3 (D3) and 25-hydroxycholecalciferol (25OHD3) in the diet
Answer:
The relevant changes were applied into the Table 1 and Table 2.
